# Hypophysitis: Defining Histopathologic Variants and a Review of Emerging Clinical Causative Entities

**DOI:** 10.3390/ijms24065917

**Published:** 2023-03-21

**Authors:** Cameron A. Rawanduzy, Alexander Winkler-Schwartz, William T. Couldwell

**Affiliations:** Department of Neurosurgery, Clinical Neurosciences Center, University of Utah, Salt Lake City, UT 84112, USA

**Keywords:** pituitary gland, hypophysitis, pituitary, adrenocorticotrophic hormone, immune checkpoint inhibitors

## Abstract

Inflammatory disease of the pituitary gland is known as hypophysitis. There are multiple histological subtypes, the most common being lymphocytic, and the pathogenesis is variable and diverse. Hypophysitis can be primary and idiopathic or autoimmune related, or secondary to local lesions, systemic disease, medications, and more. Although hypophysitis was previously accepted as an exceedingly rare diagnosis, a greater understanding of the disease process and new insights into possible etiologic sources have contributed to an increased frequency of recognition. This review provides an overview of hypophysitis, its causes, and detection strategies and management.

## 1. Introduction

The pituitary gland sits in the sella turcica, where it fulfills its role as an intricate regulator of the endocrine system. It is susceptible to a wide variety of pathologies that can affect the anterior and posterior lobes, and damage to the gland can lead to mortality and morbidity [1]. Hypophysitis is defined as general inflammation of the pituitary gland, the infundibulum, and the hypothalamus [2,3]. It is a rare condition with diverse etiologies that leads to hypopituitarism via sellar and parasellar compression and the destruction of glandular tissue. Primary hypophysitis is the most common form (Table 1), and it is often idiopathic, but our current understanding suggests that there likely is an autoimmune-mediated mechanism of inception, due to T cell-mediated recognition of autoantigens in the pituitary cells and ensuing cytotoxicity [4,5,6]. Additionally, infectious, neoplastic, or infiltrative processes can be inciting events for primary hypophysitis [3]. Secondary causes of hypophysitis include disseminated tuberculosis infection, sarcoidosis, syphilis, connective tissue disorders, Takayasu’s arteritis, Crohn’s disease, inflammation secondary to pituitary adenoma, local germinoma, craniopharyngioma or Rathke cleft cysts, Langerhans cell histiocytosis, and Wegener’s granulomatosis, among others (Table 2) [6,7,8,9]. Chronic inflammation of the pituitary gland from overactive lymphocytes damages the gland and causes pituitary atrophy and fibrosis, which ultimately impair pituitary function [10]. An empty sella can form, and patients will require long-term hormone replacement for one or more pituitary axes [4,11].

Primary hypophysitis is further classified by its anatomical location and histopathological features [2,12]. The multiple subtypes are correlated with different underlying causes, presentations, and rates of incidence. Inflammation confined to the anterior pituitary, inflammation of the infundibulum and posterior pituitary, and inflammation affecting the whole pituitary are referred to as adenohypophysitis, infundibuloneurohypophysitis, and panhypophysitis, respectively [3,13,14]. The histologic subtypes of primary hypophysitis are lymphocytic, granulomatous, xanthomatous, IgG4-related disease, and necrotizing hypophysitis [4,7,13,15]. The most common form is lymphocytic, followed by granulomatous hypophysitis [2,16]. Hypophysitis can radiologically mimic other nonfunctioning masses [17]. There is a spectrum of the symptomatic manifestations; hypophysitis can remain asymptomatic, cause mild endocrine deficiencies and mass effect, or result in major endocrine deficiencies and death [5,17]. Clinically, infundibuloneuro- and panhypophysitis present with diabetes insipidus (DI) and antidiuretic hormone deficiency, whereas anterior pituitary involvement alone typically does not result in DI; rather, it can result in anterior pituitary hormone deficiencies [2]. Although there are challenges to diagnosis and it requires a high degree of suspicion to avoid unnecessary surgery, hypophysitis should be properly differentiated from other sellar and parasellar masses [8]. Tissue biopsy is the gold standard; however, magnetic resonance imaging (MRI) is most often used to support a definitive diagnosis. Common features include infundibular thickening and the absence of cavernous sinus invasion, especially within the context of new-onset DI or an onset with a close temporal relationship to pregnancy [17]. Several antibodies to pituitary antigens have been identified as possible serum markers of hypophysitis, including those against growth hormone, alpha enolase, secretograinin-2, pituitary gland-specific factors 1a and 2, TPIT, PIT1, and chorionic somatomammotropin [5]. Presently, the prognostic utility of these antibodies and pituitary antigens is low.

## 2. Epidemiology

The earliest described case of hypophysitis is from 1962. Goudie and Pinkerton presented a 22-year-old postpartum woman who died of circulatory shock with hypothyroidism and amenorrhea [14,18,19]. Although this is the earliest credited description, it is likely that some of Sheehan’s original patient series from the early 1900s included patients with the diagnosis [14]. Historically, almost all cases were delineated as idiopathic because the underlying pathophysiologic mechanisms were unknown [5]. The most commonly referenced incidence of this disorder is 1 in 9 million persons annually, however, this estimate comes from a single-center study conducted over 2 decades ago that only accounted for lymphocytic and granulocytic forms of hypophysitis [3,5,20]. It is likely this is an underestimation. The true prevalence and incidence of hypophysitis has grown exponentially over past decade in part due to increased awareness of the condition, better imaging, and increased reporting of IgG4- and immune checkpoint inhibitor-related hypophysitis [3,5]. The average age of presentation is 31 years for women and 42 years for men. The duration of symptoms is between 1 month and 5 years, and the onset of hypopituitarism is more rapid in lymphocytic hypophysitis versus adenomas [19].

## 3. Subvariants

### 3.1. Primary Hyophysitis

#### 3.1.1. Lymphocytic

The lymphocytic subtype represents more than 70% of all cases of hypophysitis and has a well-established pathogenesis [5]. It is more common in women, with most cases reported during late pregnancy or the early postpartum period. It mainly affects the anterior pituitary [5]. Adrenocorticotrophic hormone (ACTH) deficiency is common [16]. The proposed relationship between hypophysitis and pregnancy is related to pituitary hyperplasia during the gestational period increasing pituitary antigens and alterations in hypophyseal blood flow. This results in increased pituitary blood supply and greater exposure to the immune system [5]. Lymphocytic panhypophysitis and infundibuloneurohypophysitis do not have the same sex distribution and relationship to pregnancy. Cases of pediatric patients with lymphocytic hypophysitis have been documented; these patients can present with DI and amenorrhea [21].

Lymphocytic hypophysitis usually results in a solid enlargement of the pituitary gland; cystic components are less likely [18]. Histopathologically, there is extensive infiltration of lymphocytes, predominantly T cells, into the interstitium, with additional B cells, plasma cells, and occasional eosinophils, macrophages, histiocytes, and masts cells present [14]. If left untreated, the result is fibrosis. On autopsy, patients with lymphocytic hypophysitis demonstrate a significantly atrophied, fibrous pituitary gland adhering to the dura with secondary atrophy of the adrenal glands [18,21,22].

Lymphocytic hypophysitis is associated with other autoimmune diseases in approximately 20% of cases. These may include Hashimoto thyroiditis, diabetes mellitus type 1, hypoparathyroidism, Graves disease, Addison disease, vitiligo, pernicious anemia, alopecia, myasthenia gravis, primary biliary cirrhosis, chronic atrophic gastritis, and non-organ-specific conditions, such as systemic lupus erythematosus [9]. A Rathke cleft cyst is an uncommon sellar lesion of Rathke pouch origin. Usually, these lesions remain asymptomatic, but they can rupture and cause secondary lymphocytic hypophysitis, so it is important to consider lymphocytic hypophysitis after the rupture of a Rathke cleft cyst when doing diagnostic imaging [23]. Serum anti-pituitary antibodies might be present, but they lack specificity and are present in other autoimmune disorders that have no pituitary involvement at all [3].

#### 3.1.2. Granulomatous

Granulomatous hypophysitis is the second most common variant and is characterized by multinucleated giant cells with necrotizing granulomas and peripheral histiocytes [5,15]. Overall, it is a rare diagnosis and mimics pituitary adenomas clinically and radiologically [7]. It can present idiopathically, or secondary to infection, sarcoidosis, Crohn’s disease, Rathke cleft cyst rupture (Figure 1), or pituitary adenoma [15]. It represents approximately 20% of all cases of panhypophysitis [5], and while there is a female sex predominance, it is less than the sex predisposition of lymphocytic hypophysitis. This subvariant presents in the 5th decade of life. It is not associated with pregnancy [5,15,22].

Granulomatous hypophysitis may be more clinically severe than lymphocytic hypophysitis with a higher frequency of headaches and visual disturbances due to chiasmal compression, and hypopituitarism causing endocrine dysfunction and DI in up to 75% of patients [4,15,22].

#### 3.1.3. Xanthomatous

Xanthomatous hypophysitis is rare and was first reported in 1998 by Folkerth [6]. It is commonly cystic, and this histopathology is defined by its lipid-rich foamy histiocytes [6,18,24,25]. Xanthomatous hypophysitis accounts for 3% of panhypophysitis cases, and it is more common in women than men, with a 3:1 female-to-male ratio [5]. This subtype has similar features to nonfunctional adenomas, and diagnosis is not possible without a biopsy and pathology report [6].

The pathogenesis is commonly due to an inflammatory response to components of Rathke cleft cyst that are exposed after rupture, hemorrhage, or leakage [4,25,26]. The initial precipitating event causes a secondary response to the mucoid fluid content released from the cyst, and the resulting inflammation becomes a chronic process, recruits macrophages, and accumulates lipids [24,25]. Xanthomatous and xanthogranulomatous hypophysitis represent a spectrum with clinicopathologic overlap. Xanthogranulomatous lesions contain cholesterol clefts, fibrosis, giant cells, eosinophilic necrotic debris, macrophage accumulation, and hemosiderin deposits, whereas there is no hemosiderin pigment in xanthomatous lesions. This suggests the xanthomatous form can transition to xanthogranulomatous possibly as a chronic process through repeated Rathke cleft cyst hemorrhages [26].

Grossly, xanthomatous hypophysitis causes a pituitary cyst filled with orange/amber-colored thick fluid and pus-like floating crystals. It can affect the posterior pituitary and hypothalamus, but there is usually only anterior involvement [24]. Histologically, there is a mixed inflammatory infiltrate composed of lipid-laden histiocytes that look like xanthomas. It is nonneoplastic, and it behaves more like a proliferative mass than lymphocytic or granulomatous hypophysitis, eliciting symptoms from mass effect [6,9]. On imaging, the lesions display more symmetric enlargement of the pituitary gland than adenomas [24]. Radical resection is required to remove the cyst, but recurrence is not uncommon [26].

#### 3.1.4. IgG4-Related Disease

IgG4-related hypophysitis is a novel, emerging cause for pituitary inflammation that typically is part of an overarching systemic disease process affecting multiple organs. The pathogenesis is not well described, but there is a reported association with BRAF V600E mutations [12]. This subtype affects males more frequently than females, and there are various degrees of involvement of the anterior and posterior pituitary and the stalk [4,5]. Histologically, it is defined by fibrosclerotic systemic disease with dense infiltration of IgG4-positive plasma cells into the pituitary gland. A thickened stalk can be present, and the diffuse infiltrate notably leads to a storiform pattern of fibrosis [12]. It is often associated with inflammatory disease in the salivary glands, pancreas, or with retroperitoneal fibrosis. Infrequently, IgG4 causes isolated pituitary disease; however this only occurs in 4–5% of cases [9,27]. The clinical presentation of IgG4 hypophysitis is similar to that seen with immune checkpoint inhibitor-related hypophysitis [27]. Elevated serum IgG4 is not diagnostic. Patients can present with nonspecific, acute symptoms that mimic pituitary apoplexy with normal serum IgG4 levels. Prompt treatment is necessary to alleviate symptoms [12].

#### 3.1.5. Necrotizing

Necrotizing hypophysitis is reportedly the rarest histologic variant; it represents approximately 0.6% of all cases of panhypophysitis, there are few reported cases in the literature, and the pathogenesis is not well elucidated [4,5]. It is defined by extensive areas of necrosis in the pituitary from infiltration of lymphocytes, plasmacytes, and eosinophils. Similar to the more common subtypes, there is a female predominance. Necrotizing hypophysitis may be more likely to present with acute mass effect [5].

### 3.2. Secondary Causes

#### 3.2.1. Immunotherapy

The first reported case of immune checkpoint inhibitor (ICI)-induced hypophysitis is from the National Cancer Institute in 2003, in a patient with metastatic melanoma managed with ipilimumab, the monoclonal antibody for CTLA4 [10]. The growing popularity of novel cancer treatments has led to an increase in adverse immune-related events including hypophysitis [4]. Immunotherapeutic drugs can cause hypophysitis and functional deficits in one or more pituitary axes with subtle or no changes on MRI [10]. ICIs can treat malignant tumors as a standalone therapy or an adjunct to surgery, radiosurgery, chemotherapy. [28] Two targets of ICI are CTLA4 and PD1. CTLA4 is a member of the immunoglobulin family expressed in regulatory T cells that resides on the cell surface, and its activation increases immune cell proliferation and production of immunosuppressive cytokines [10]. PD1 is a member of the immunoglobulin family. It is expressed on effector T cells, and it attenuates T cell function. [10] These monoclonal antibody treatments modulate the immune system, augment the T cell response, and prevent cancer invasion in difficult-to-treat tumors [29,30]. The ensuing immune-related adverse events are driven by cytokine dysregulation [29].

ICI hypophysitis is more common in males and occurs at an older age than other causes of hypophysitis, although this may reflect the demographics of ICI treatment or underlying tumors [10] Hypophysitis occurs relatively frequently in patients treated using the CTLA4 blocker ipilimumab; the incidence of ICI is nearly 12%. In contrast, the PD1 monoclonal antibodies camrelizumab and pembrolizumab cause hypophysitis in 0.5% of patients [31]. Hypophysitis occurs in less than 0.1% of patients treated with PDL1 inhibitors [4,27,30,32]. Combination therapy with CTLA4 and PD1 inhibitors causes hypophysitis in 8.8 to 10.5% of patients [30].

The pathogenesis of anti-CTLA-4 antibody-induced hypophysitis is a result of type II and IV hypersensitivity and the humoral immune response [4]. It is possible that ectopic expression of CTLA4 on prolactin and thyroid-stimulating hormone (TSH)-secreting cells in the pituitary results in development of anti-pituitary antibodies [10,12,30]. Other possible mechanisms are ICI-induced expanded T cell populations that release pro-inflammatory cytokines and epitope spreading secondary to tumor lysis [10]. In monoclonal antibodies towards PD1/PDL1, there is no clear pathogenesis delineated [10,12,30], although it may due to specific antibodies against ACTH from ectopic expression of ACTH in a tumor leading to autoimmunity against corticotropic cells and a paraneoplastic syndrome [30].

Headache is present in 80–85% of patients with CTLA4 hypophysitis, and the majority display pituitary and multiple hormonal deficiencies. Hypophysitis can occur 6–11 weeks after beginning treatment. In anti-PD1/PDL1 ICI, hormone deficiencies are usually isolated to ACTH, and hypophysitis appears later during the course of treatment [30]. In contrast to anti-CTLA4 monoclonal antibodies, hypophysitis from anti-PD1 monoclonal antibodies appears after 6 months [27]. Patients present with fatigue, nausea, and loss of appetite [27,30]. A rare reported presentation was seen in a patient with unexpected fatigue, nausea, hoarse voice, bucking, and difficulty in breathing [28]. Because symptoms of hypophysitis can be nonspecific and also related to a cancer diagnosis, clinical suspicion and awareness of the risk are imperative [27,30].

#### 3.2.2. COVID-19

Pituitary apoplexy is a known rare phenomenon that can occur after infection with COVID-19. Additionally, there are several case reports of DI that have been described in patients weeks after COVID-19 infection [33]. Recently, there have been reports in the literature of hypophysitis related to the SARS-CoV-2 virus as well as the vaccine. Ankireddypalli et al. [34] reported a woman who experienced severe, persistent headache, myalgias, polyuria, and polydipsia two days after COVID-19 vaccination. MRI was obtained, and a thickened pituitary stalk was identified. The patient was diagnosed with DI and hypophysitis. Symptoms improved on 3-month follow-up, however, the thickened pituitary stalk remained. Additionally, there are reports of hypophysitis occurring between 2 weeks and 2 months after COVID-19 infection. Patients began experiencing headaches, laboratory-identified hypothyroidism, hypocortisolism, hypogonadism, and MRI supported a diagnosis of hypophysitis [35,36]. Although this is a rare complication, signs and symptoms of hypophysitis after a recent COVID-19 infection or vaccine should warrant appropriate investigation.

#### 3.2.3. Paraneoplastic Syndrome

Hypophysitis caused by a paraneoplastic autoimmune reaction is another novel clinical entity. The presence of various ectopic proteins expressed in tumors, such as pituitary transcription factors or hormones, can lead to the formation of autoantibodies and autoreactive cytotoxic T cells [37]. Pit-1 is a transcription factor required for the differentiation, proliferation, and maintenance of pituitary somatotrophs, lactotrophs, and thyrotrophs [12]. Primary tumors like thymoma and bladder lymphoma can cause a paraneoplastic syndrome [5,37], and anti-Pit1-hypophysitis will lead to growth hormone, prolactin, and TSH deficiencies with detectable circulating anti-PIT1 antibodies [12]. A proposed diagnostic algorithm from Yamamoto et al. [38] suggests three criteria to identify anti-Pit1-hypophysitis: acquired growth hormone, TSH, and prolactin deficiencies without impairment of other pituitary axes; presence of anti-PIT1 antibodies or PIT1-reactive cytotoxic T-lymphocytes; and an underlying diagnosis of thymoma or malignant neoplasm [5,37].

#### 3.2.4. Autoimmune and Local Infiltrative Disease

Primary hypophysitis is associated with another autoimmune disease in approximately 25–50% of cases. The most common autoimmune diseases that appear in concert with hypophysitis include Hashimoto thyroiditis, Graves disease, Addison disease, diabetes mellitus type 1, atrophic gastritis, systemic lupus erythematosus, Sjogren syndrome, primary biliary cirrhosis, Erdheim—Chester disease, and autoimmune hepatitis [14]. Immune processes likely target specific pituitary cell subtypes, and loss of ACTH-, follicle stimulating hormone/luteinizing hormone-, or TSH-secreting cells is seen [14].

Hypophysitis secondary to pituitary abscess is rare, but can be life threatening [11]. Most commonly, these cases are of bacterial origin; however, rarely, fungal infections, such as aspergillosis and coccidioidomycosis can affect the pituitary. Abscesses can cause anterior pituitary hormone deficiencies or central DI. Growth hormone deficiency is the earliest manifestation, followed by deficient gonadotropic hormones, then TSH and ACTH deficiencies [11].

Tuberculosis is a cause of secondary granulomatous infectious hypophysitis in the developing world. It can cause anterior pituitary dysfunction, such as hypopituitarism and hyperprolactinemia with galactorrhea and amenorrhea. DI, polyuria and polydipsia are commonly observed [11].

Other granulomatous pituitary lesions include sarcoidosis, idiopathic giant cell granulomatous, Takayasu arteritis, Crohn’s disease, and Cogan syndrome [4,11]. Thus, in a patient presenting with signs of a pituitary lesion, a thorough history should be obtained. In a patient with known history of systemic disease, signs of hormone dysregulation, and/or headaches and visual disturbances, hypophysitis should be considered in a differential diagnosis [11].

## 4. Diagnosis

### 4.1. Imaging Features

If a clinical picture raises suspicion of possible hypophysitis and pituitary dysfunction, a full pituitary hormone panel and imaging should be obtained [20]. Clinical and radiological assessment and confirmation of hypophysitis is challenging, but with advancements in imaging technology and sophistication, the recognition of hypophysitis cases is increasing (Table 3) [3].

MRI is the imaging modality of choice, but diverse features can appear on presentation. Patients with autoimmune hypophysitis are regularly misdiagnosed as having pituitary macroadenoma and can undergo unnecessary surgery [22]. To avoid this, primary hypophysitis must be differentiated from other sellar and parasellar masses [4]. MRI features that are more indicative of autoimmune hypophysitis include a symmetric enlargement of the pituitary gland, intense homogenous enhancement after gadolinium administration, a thickened non-tapering stalk, and an intact sellar floor (Figure 2) [39,40]. In lymphocytic hypophysitis, additional dark signal intensity on T2-weighted MRI, including the parasellar T2 dark sign, has been reported as a characteristic finding [41]. In contrast, pituitary macroadenomas are asymmetric, displace the infundibulum, and rarely involve the stalk or erode the sellar floor [4,20,22,40]. Additional MRI findings that can be correlated with clinical features include a lower pituitary volume, loss of the posterior pituitary bright spot, absence of mucosal thickening, or a triangular or dumbbell-shaped gland, in a patient whose age is less than 30 years or who is a pregnant or postpartum female [8,20,42].

Computed tomography scan can sometimes feature adjacent bony changes. In IgG4-related disease, isohypointensity on T1- and T2-weighted imaging can be present and correspond to the degree of fibrosis and increased cellular infiltration [43]. However, a gadolinium-enhanced MRI is necessary for sellar and parasellar visualization [8]. The presence of pituitary stalk thickening is the strongest predictor of hypophysitis. The other most significant findings are diffuse, mild to moderate pituitary expansion with significant homogenous or heterogeneous gadolinium enhancement [8,18,27].

Gutenberg et al. [22] created a clinicoradiological score based on relation to pregnancy, pituitary volume and symmetry, signal intensity and homogeneity after gadolinium administration, posterior pituitary bright spot, stalk size, and mucosal swelling. Within this rating system, symptoms appearing during pregnancy or the postpartum period favored hypophysitis. Volume was significantly greater in adenomas—volumes greater than 6 cm^3^ were more likely to be an adenoma than hypophysitis. Gadolinium uptake was higher in hypophysitis, and the presence of asymmetric expansion favored a diagnosis of adenoma. Finally, the posterior pituitary bright spot was absent more frequently in hypophysitis. Up to 48% of the population normally has an absent posterior pituitary bright spot, so it is not pathognomonic for hypophysitis [8].

In ICI-induced hypophysitis, imaging changes can precede onset of symptoms. One study reported that 50% of patients presented with diffuse pituitary enlargement on MRI before any clinical manifestations of disease [4]. Pituitary enlargement and visual disturbance are more common in primary hypophysitis than ICI-induced hypophysitis [12]. ICI-induced hypophysitis is associated with the disappearance of precontrast T1-weighted hyperintensity in the posterior aspect of pituitary on MRI and variable stalk enlargement (Figure 3). The pituitary may appear homogenously hyperintense on T1-weighted postcontrast sequences. The use of anti-CTLA4 monoclonal antibodies results in identifiable MRI abnormalities more frequently than PD1 inhibitors [32]. In patients prescribed anti-CTLA4 ICI, it is reasonable to get a pituitary MRI at baseline [44] and monitor for hormonal or mass effect inducible changes in the first 6 months of treatment. Additionally, although the MRI may appear normal in a patient on anti-PD1/PDL1 therapy with clinical signs of hypophysitis, obtaining the imaging is important in order to rule out pituitary metastases [30].

In xanthomatous hypophysitis, specifically, an MRI can identify a cystic, enlarged pituitary gland, occasional stalk thickening, and cavernous sinus invasion that can lead to a characteristic triangle saddle occupation on coronal plane MRI [6].

### 4.2. Symptomatology

Hypophysitis can present as acute, subacute, or chronic [14]. Many symptoms of hypophysitis also occur in apoplexy and tumors, but sudden-onset endocrine deficiencies and presence of DI are more suggestive of nonadenomatous sellar lesions [11,20]. Pituitary tumors tend to cause isolated growth hormone or gonadotrophic dysfunction, whereas isolated ACTH or TSH deficit more likely has an inflammatory and autoimmune cause [14]. Endocrine symptoms or neurological symptoms due to mass effect are common [20]. These include headache, nausea, visual disturbances, diplopia, cranial nerve 3, 4, and 6 neuropathies, hydrocephalus, and mental status changes [11,46]. Carotid artery occlusion and symptoms that mimic meningitis or pituitary apoplexy are consequences of extensive mass effect [2,14,46].

Hypopituitarism and central DI are the most clinically significant problems and require immediate attention [7]. Lymphocytic hypophysitis has a predilection to dysregulate ACTH, gonadotrophins, TSH, then growth hormone, in that sequence, and isolated hormone deficits are rare [2,46]. This is contrary to other causes of hypopituitarism, where impaired initial ACTH or TSH secretion is uncommon. [4]. The explanation of selectivity for precocious ACTH deficiency compared with pituitary adenoma is not well elucidated.

Acute presentations are similar to that of a nonsecretory pituitary tumor with ACTH and secondary adrenal insufficiency. With rapid onset, hypocortisolism is associated with a high rate of mortality and morbidity and can result in sudden death without early recognition and management [14,27]. Chronic hypophysitis leads to pituitary atrophy and hormone insufficiency, so patients will require lifelong hormone replacement therapy [23].

Subacute onset of symptoms is seen in young women who are towards the end of pregnancy or recently postpartum. In pregnant women, the differential diagnosis for headaches includes pre-eclampsia, post-dural puncture headache, central venous sinus thrombosis, and Sheehan syndrome [14,46]. Hypophysitis is an important consideration. In these patients, laboratory tests for urinalysis and hyponatremia are recommended.

In ICI-related hypophysitis, DI is rare [12,47]. Polyuria and polydipsia, visual disturbances, hyperprolactinemia, and hypogonadism are less frequently observed due to the milder, transient enlargement of the pituitary gland [10]. Patients taking ICI who develop hypophysitis will almost invariably have ACTH deficiency, and fatal, acute adrenal insufficiency has been reported [4,10]. TSH deficiency is also very common. Symptoms in ICI-related compared to primary hypophysitis are often related to decreased cortisol, including fatigue, appetite loss, nausea, vomiting, malaise, dizziness, and mild cognitive defects. They can be subtle or life threatening [10], which highlights the importance of pituitary function tests at baseline and monitoring during treatment. The French Society of Endocrinology guidelines for patients starting ICI include evaluating fasting, venous glycemia, natremia, TSH, free T4, 8am cortisol, and, additionally, testosterone in men during the first six months of treatment at each course of ICI and then every two courses during the following six months [30]. Glucocorticoid replacement should be started without delay if there is strong suspicion of adrenal insufficiency [4,10].

In cancer patients, similar symptoms of headache, DI, and hypopituitarism can also present in pituitary metastases, and it is important to distinguish between the two diagnoses because management will be different. In a study of DI in pituitary metastases versus autoimmune hypophysitis and immune checkpoint inhibitors, DI was significantly more common in the pituitary tumors [10].

### 4.3. Management

As previously stated, the initial approach to suspected primary hypophysitis encompasses prompt endocrine workup and evaluation of the pituitary gland-related hormonal profile, including cortisol, ACTH, insulin-growth factor-1, growth hormone, estradiol/testosterone, follicle stimulating hormone/luteinizing hormone, free thyroxine, TSH, prolactin, and plasma/urine osmolality and electrolyte [12,13].

Primary hypophysitis can be self-limited, and spontaneous remission may occur in some instances [2]. Otherwise, treatment options include surgery, medical management, observation, or a combination of these therapies [23]. Conservative management is recommended unless symptoms are severe and progressive. The only exception is IgG4-related hypophysitis, which should be promptly treated to revert symptoms and prevent irreversible fibrosis [4]. Patients with systemic IgG4-related disease and multiple organ involvement may benefit from long-term steroid therapy [12].

The utility of surgery is the subject of debate in the literature, but it is universally agreed that decompressive surgery should be performed if there is significant and dangerous mass effect [2]. Surgery will not resolve endocrine dysfunction related to destruction of pituitary hormone-secreting cells, but pituitary surgery can relieve symptoms of hypogonadism and hyperprolactinemia secondary to glandular compression [14]. Additionally, because of the necessity of histopathologic assessment, surgical biopsy is the only method of obtaining a definitive diagnosis; therefore, surgery may be warranted if neoplasm or another pathology cannot be ruled out from the clinical picture and imaging [14,19,23]. If a biopsy would change therapeutic management, it should be considered [22].

The mainstay treatment for hypophysitis is glucocorticoids [2,20]. In a systematic review, the use of glucocorticoids in panhypophysitis yielded a better improvement in visual field and corticotroph axis recovery with intravenous route delivery, very-high-dose glucocorticoids, and a duration of longer than 6.5 weeks. High and very high doses, >100 mg/day, resulted in better outcomes than moderate doses of glucocorticoids [48].

An acute phase presentation of hypophysitis with mass effect causing progressive headaches is an indication for treatment. The presence of compression should be confirmed on MRI, after which high-dose glucocorticoids can be administered [12]. The resolution of neuroradiological abnormalities is usually within 2 months. Treatment with high-dose glucocorticoids will not restore ACTH deficiency, and most patients require long-term replacement. Thyroid and gonadal deficiencies often recover, and the need for hormone replacement must be reassessed in the long term [4]. Chronic hypophysitis and pituitary atrophy will require hormone replacement therapy [2,5].

Pure steroid therapy may be less effective in granulomatous than lymphocytic hypophysitis [7]. In xanthomatous hypophysitis, the benefit of high-dose glucocorticoids is controversial [26]. In a case report by DeCou et al. [26], xanthogranulomatous hypophysitis secondary to a ruptured Rathke cleft cyst was initially treated with dexamethasone, then administered a maintenance course of celecoxib and mycophenolate mofetil. Steroid-sparing immunosuppressive therapy in combination with an anti-inflammatory can be a safe and effective treatment for recurrent hypophysitis. Additionally, other steroid-sparing therapies like azathioprine have been used. Methotrexate, cyclosporine, infliximab, gamma knife, stereotactic radiosurgery, and rituximab are other options that have successfully treated hypophysitis [14,20,26]. Rituximab might be an effective alternative treatment in B cell-predominant autoimmune hypophysitis [49].

## 5. Conclusions

Hypophysitis is an inflammatory disorder of the pituitary gland with variable etiologies that can disrupt multiple hormonal axes and have dire consequences. Although hypophysitis was previously thought to be an exceedingly rare phenomena, a more accurate definition and greater understanding of the pathogenesis has contributed to the knowledge that it may occur more frequently than anticipated. Moreover, with the growth in the popularity of ICIs for malignancy, and the post-viral and post-vaccine-related hypophysitis associated with COVID-19, the incidence could be increasing still. Looking to the future, a continued search for reliable markers and/or pituitary autoantibodies is necessary to develop accurate and noninvasive, diagnostic tests.

## Figures and Tables

**Figure 1 ijms-24-05917-f001:**
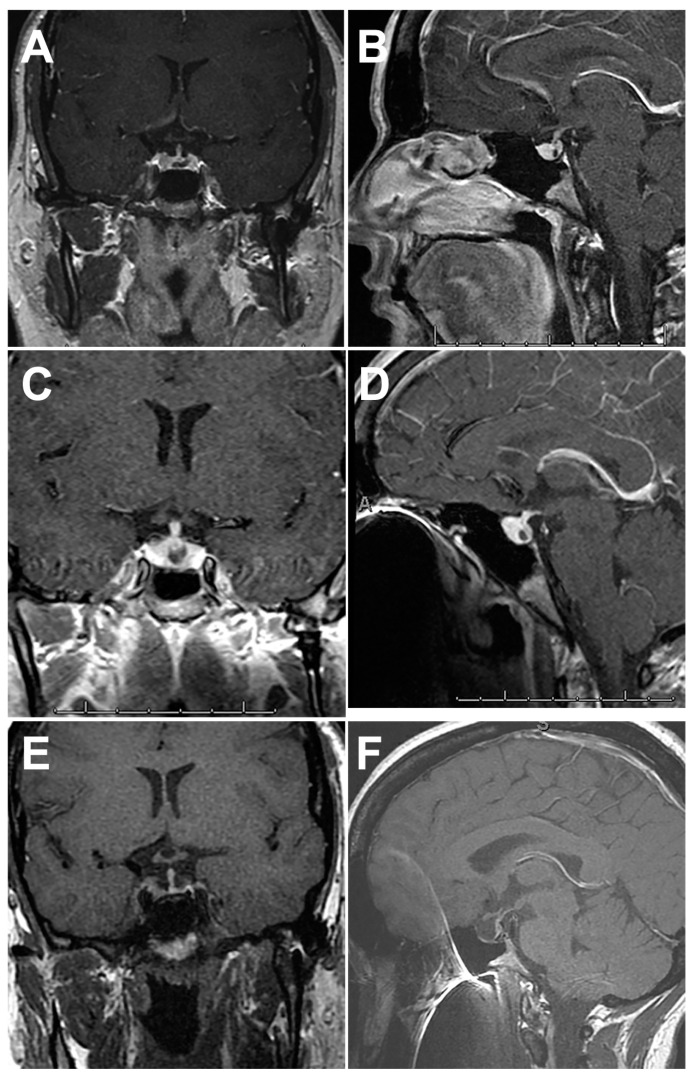
This 34-year-old woman presented with a 2-year history of galactorrhea and hyperprolactinemia, treated successfully with cabergoline. During previous treatment for Crohn disease, gadolinium-enhanced MRI showed a normal-appearing pituitary gland with Rathke cleft cyst (RCC) in the intermediate lobe ((**A**), axial; (**B**), sagittal). At 2-year follow-up, she had progressive headaches and her menstrual periods had ceased. MRI demonstrated the RCC and definite enlargement of the pituitary ((**C**), axial; (**D**), sagittal). No definitive adenoma was identifiable. Endocrine labs demonstrated normal prolactin, low gonadotroph and estrogen levels (central hypogonadism), and low normal thyroid hormone (all consistent with partial hypopituitarism). She underwent pituitary biopsy and drainage of the RCC. Pathological evaluation demonstrated ciliated low cuboidal epithelium and surrounding chronic lymphocytic inflammation. The anterior pituitary gland had marked mononuclear inflammation. The diagnosis was non-necrotizing granulomatous hypophysitis. She was treated with 40 mg/0.8 mL adalimumab weekly. Her headaches resolved but she developed progressive panhypopituitarism requiring replacement therapy. MRI demonstrated progressive pituitary atrophy 2 years later ((**E**), axial; (**F**), sagittal).

**Figure 2 ijms-24-05917-f002:**
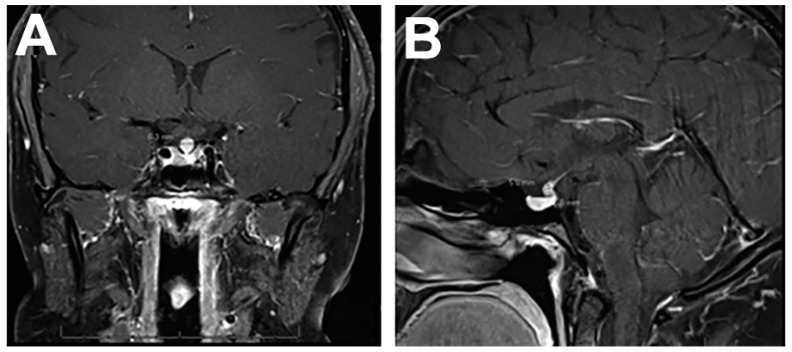
Coronal (**A**) and sagittal (**B**) T1-weighted MRI of a patient who initially presented with diabetes insipidus, showing focal nodular thickening and enhancement of the pituitary infundibulum with absence of neurohypophysis T1 signal. Biopsy and pathology found T cell predominant mixed inflammatory infiltrate. The clinicoradiologic picture was consistent with lymphocytic hypophysitis.

**Figure 3 ijms-24-05917-f003:**
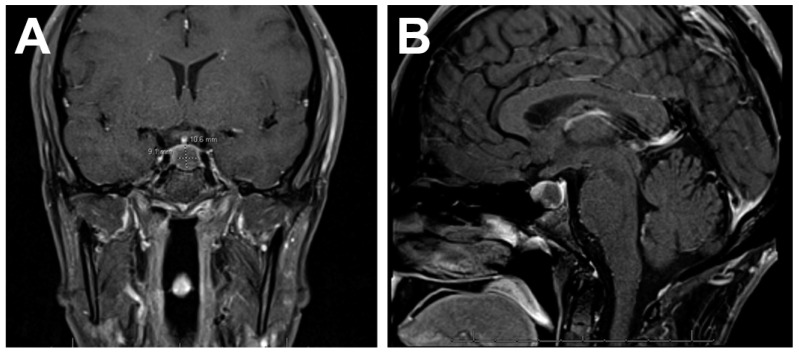
Coronal (**A**) and sagittal (**B**) T1-weighted MRI of a patient being treated with pembrolizumab that began experiencing signs of pituitary apoplexy with headache and extreme fatigue. T1 hypointense lesion is visible in the posterior left aspect of the pituitary gland, which is in the region of the previous T1 hyperintense posterior pituitary bright spot. Pathological analysis showed fragments of anterior and posterior pituitary gland with marked mononuclear inflammation and CD3+ T cell infiltrate. Reprinted with permission from Ref. [45] Copyright 2021. John Wiley & Sons.

**Table 1 ijms-24-05917-t001:** Primary histopathologic subtypes of hypophysitis.

Subtype	Characteristics
Lymphocytic	⋅ Most common histopathologic subtype seen in pregnancy. ⋅ Defined predominantly by interstitial T cells, additional B cells, plasma cells, and other immune cells.
Granulomatous	⋅ Second most common primary subtype. ⋅ Multinucleated giant cells with granulomas and histiocytes are visualized on histopathology. ⋅ Radiologically, can appear similar to pituitary adenomas.
Xanthomatous	⋅ A rare, cystic form of hypophysitis characterized by its lipid-rich, foamy histiocytes on histopathology. ⋅ Commonly occurs due to an inflammatory response to ruptured or hemorrhaged Rathke cleft cysts. This presentation has brought into question whether xanthomatous should be designated as a primary or secondary disease incidence.
Ig-G4-related disease	⋅ Isolated IgG4-related hypophysitis is possible, but it is rare and usually occurs as part of a multi-organ-affected systemic disease. ⋅ Histopathologically, fibrosclerosis, dense infiltration of IgG4-positive plasma cells, and a diffuse, storiform pattern of fibrosis in the pituitary stalk are visualized.
Necrotizing	⋅ Frequently reported as the rarest histopathologic subvariant. ⋅ Extensive areas of necrosis with lymphocytes, plasmacytes, and eosinophils can be visualized. ⋅ Can cause acute mass effect.

**Table 2 ijms-24-05917-t002:** Secondary autoimmune and emerging, novel clinical causes of hypophysitis.

Subtypes	Characteristic
ICI-induced	⋅ The inception and growing frequency of cancer patients (lung cancer, melanoma) being treated with immune checkpoint inhibitors has led to increased incidence of hypophysitis. ⋅ CTLA4 inhibitors can cause hypophysitis in rates upwards of 15%. ⋅ It is prevalent, but less common, in patients on PD1/PDL1 inhibitors.
COVID-19	⋅ The continued aftermath of the COVID-19 pandemic has resulted in case reports of post-viral and possible vaccine-related adverse side effects. ⋅ Although rare, documented cases of hypophysitis recently after infection by or vaccination against SARS-Cov-2 warrant further investigation.
Paraneoplastic	⋅ Autoimmune destruction of the pituitary gland can occur in patients with formation of anti-Pit-1 antibodies. ⋅ Ectopic tumor expression of Pit-1, a transcription factor for somatotrophs, lactotrophs, and thyrotrophs, can lead to hypophysitis in patients with thymoma and bladder lymphoma.
Autoimmune	⋅ Hypophysitis is frequently associated with additional autoimmune diseases. ⋅ Most commonly, past medical history can include Hashimoto thyroiditis, Graves’ disease, Addison disease, diabetes mellitus type 1, atrophic gastritis, systemic lupus erythematosus, Sjogren syndrome, Erdheim—Chester disease, and autoimmune hepatitis among others.
Granulomas and infection	⋅ An uncommon manifestation of infectious processes is hypophysitis. ⋅ Local disease infiltrating into the pituitary can cause inflammatory-related destruction of hormone-secreting cells. ⋅ Possible culprits include bacterial infection, such as tuberculosis, and fungal abscesses due to aspergillus or coccidiomycosis. ⋅ In addition to tuberculosis, other granuloma-defined diseases can cause pituitary lesions, including sarcoidosis, idiopathic giant cell granulomatosis, Takayasu arteritis, Crohn’s disease, and Cogan syndrome.

**Table 3 ijms-24-05917-t003:** Differentiation among common clinical and imaging features and management in hypophysitis versus pituitary adenoma.

Features	Adenoma	Hypophysitis
Clinical Symptoms	⋅ Bitemporal hemianopia, headache, hormonal deficiencies. ⋅ Most commonly, adenomas are nonfunctioning or prolactinoma. ⋅ Can lead to gonadotrophin deficiencies, amenorrhea, galactorrhea, etc.	⋅ Sudden onset DI and endocrine deficiencies. ⋅ Isolated ACTH or TSH deficiency are more likely nonadenomatous pituitary lesions. ⋅ Headache, new-onset DI in late pregnancy or early-postpartum period should include *lymphocytic hypophysitis* in differential. ⋅ Headache, fatigue different from baseline in a patient with cancer treated with immune checkpoint inhibitors is concerning for *ICI-induced hypophysitis.*
MRI	⋅ Asymmetric pituitary enlargement, displaced pituitary stalk, sellar floor thinning, optic chiasm compression.	⋅ Symmetric enlargement of the pituitary. ⋅ Homogenous gadolinium contrast enhancement. ⋅ Infundibulum thickening. ⋅ Posterior pituitary T1 bright spot absent ⋅ Chronic inflammation may progress to an empty sella. ⋅ Imaging changes that precede symptoms are common in *ICI-induced hypophysitis*. ⋅ Diffuse pituitary enlargement, absence of posterior pituitary T1 bright spot, variable stalk enlargement. ⋅ *Xanthomatous hypophysitis* may be associated with a cystic mass, thickened stalk, and cavernous sinus invasion (i.e., triangle saddle occupation). ⋅ *Necrotic hypophysitis* does not enhance with contrast.
Management	⋅ If symptomatic, microadenomectomy to preserve vision, pituitary function, and remove tumor. ⋅ Functioning pituitary adenomas can be treated with initial medical therapies (i.e., dopamine agonists, somatostatin analogs).	⋅ Hypophysitis can be self-limited. ⋅ Rapid-onset hypocortisolism requires prompt hormone replacement. ⋅ Conservative management is first-line treatment in most cases, glucocorticoids can be administered to decrease inflammation and reverse compressive effect on surrounding sellar structures. ⋅ Acute mass effect, visual impairment, and panhypopituitarism may require decompressive surgery. ⋅ Hormone replacement therapy must be considered.

## Data Availability

No new data were created or analyzed in this study. Data sharing is not applicable to this article.

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
