# Peer review of "Hypophysitis: Defining Histopathologic Variants and a Review of Emerging Clinical Causative Entities"

_ijms, 2023, doi:10.3390/ijms24065917_

Round 1

Reviewer 1 Report

Rawanduzy et al have worked on summarizing the currently available literature on hypophysitis, an inflammatory disease of the pituitary gland. The authors do a great job describing the subtypes of primary and secondary hypophysitis. The inclusion of COVID-19 as one of the secondary causes of hypophysitis was refreshing and updated with latest literature. The pros and cons of MRI imaging were well described. The summary on disease management was comprehensive with respect to the currently available resources.

Author Response

Thank you for your positive comments.

Reviewer 2 Report

In this manuscript, the authors systematically describe and review the hypophysitis including its epidemiology, classification based on etiological subtypes, and management. Especially the hypophysitis associated with COVID-19 may attract global attentions.

This novel subject warrants a publication in IJMS, upon addressing the following minor concerns.

1.The authors should organize the manuscript into distinct sections with suitable logical titles. Such like the title 4 MRI features, maybe diagnosis as title is more appropriate. The authors should summarize the hypophysitis diagnosis including MRI features or biopsy and so on. Maybe it’s better to reorganize the content of title 4, 5 and 6.

2. about the title 3.6 Secondary Causes, maybe it’s input error. It should be 3.2. The follow-up 3.6.1-3.6.4 also should change to 3.2.1-3.2-4.

To conclude, the scientific content of the manuscript is of high quality. The selected topic is interesting and significant.

Author Response

  1. Because imaging is a key point of diagnosis and in the therapeutic decision, I believe that this chapter should be synthesized in a table so that specific MRI changes are highlighted, possible peculiarities of some histological subtypes but also diagnostic elements differential with other pathological sellar region disorders. MRI images from personal cases should be presented, if available, for better illustration and understanding. Even histological images showing the most important differences between subtypes would be informative to readers.

RESPONSE: We agree that imaging is an important component of the manuscript and have added Figures 1 and 2 to display relevant examples from cases at our institution. To your second point, we have organized the information in section 4, “Diagnosis” into a concise Table 3 to highlight the clinical, imaging, and management features as they relate to hypophysitis contrasted with pituitary adenoma, the most common lesion of the sella.

  1. A table could also summarize the clinical aspects, because in the text some information is repetitive and unclear. I would suggest a grouping of information that touches on several aspects (local inflammation expansion, onset/evolution...) I would find it useful a schema that brings together clinical, diagnostic and management data with the indication of subtypes that may have clinical or management peculiarities.

RESPONSE: As mentioned above, we have created Table 3 to synthesize relevant clinical, diagnostic, and management data. We have incorporated additional subtype-specific findings that may be present.

  1. I also suggest checking the text for possible repetitions or overlapping text (e.g. line 298). I think the bibliography can also be expanded with newer articles, especially related to imaging.

RESPONSE: Although there is some overlap in the text to demonstrate the overlapping features between hypophysitis and adenoma and within the different hypophysitis histologic subtypes, we have checked for repetitions including ~line 298. We have updated the bibliography such that two-thirds (32/48) of the references were published in just the last 3 years (2020-2023). In particular, we have added the following recent publications:

Perosevic, M., P.S. Jones, and N.A. Tritos, Magnetic resonance imaging of the hypothalamo-pituitary region. Handb Clin Neurol, 2021. 179: p. 95-112.

Lv, K., X. Cao, D.Y. Geng, and J. Zhang, Imaging findings of immunoglobin G4-related hypophysitis: A case report. World J Clin Cases, 2022. 10(26): p. 9440-9446.

Kurosaki, M., M. Sakamoto, A. Kambe, and T. Ogura, Up-To-Date Magnetic Resonance Imaging Findings for the Diagnosis of Hypothalamic and Pituitary Tumors. Yonago Acta Med, 2021. 64(2): p. 155-161.

Amereller, F., A.M. Kuppers, K. Schilbach, J. Schopohl, and S. Stormann, Clinical Characteristics of Primary Hypophysitis - A Single-Centre Series of 60 Cases. Exp Clin Endocrinol Diabetes, 2021. 129(3): p. 234-240.

Nada, A., R. Bhat, and J. Cousins, Magnetic resonance imaging criteria of immune checkpoint inhibitor-induced hypophysitis. Curr Probl Cancer, 2021. 45(1): p. 100644.

Gosangi, B., L. McIntosh, A. Keraliya, D.V.K. Irugu, A. Baheti, A. Khandelwal, R. Thomas, and M. Braschi-Amirfarzan, Imaging features of toxicities associated with immune checkpoint inhibitors. Eur J Radiol Open, 2022. 9: p. 100434.

Reviewer 3 Report

The manuscript provides a detailed description of the main histological types of hypophysitis, characterizes both imaging and clinical aspects of the different histological subtypes and discusses the principal differential diagnoses.

I appreciate the idea of this review and the way it was done, as well as the logical writing of the article. I believe, however, that there are a few points where information can be better systematized for clarification and facilitation of understanding by the reader.

So, I will point out some major revisions that would be necessary and would improve the publication.

Because imaging is a key point of diagnosis and in the therapeutic decision, I believe that this chapter should be synthesized in a table so that specific MRI changes are highlighted, possible peculiarities of some histological subtypes but also diagnostic elements differential with other pathological sellar region disorders. MRI images from personal cases should be presented, if available, for better illustration and understanding. Even histological images showing the most important differences between subtypes would be informative to readers.

A table could also summarize the clinical aspects, because in the text some information is repetitive and unclear. I would suggest a grouping of information that touches on several aspects (local inflammation expansion, onset/evolution...)

I would find it useful a schema that brings together clinical, diagnostic and management data with the indication of subtypes that may have clinical or management peculiarities.

I also suggest checking the text for possible repetitions or overlapping text (e.g. line 298)

I think the bibliography can also be expanded with newer articles, especially related to imaging.

Author Response

(The authors gave the same response as above.)

Round 2

Reviewer 3 Report

The authors have made all the changes/completions suggested during the first revision of the manuscript and I believe that the present form is suitable for publication without other changes.